# Understanding Perceived Motives for Dating Violence Among Adolescents: A Mixed-Methods Approach

**DOI:** 10.3390/bs16010031

**Published:** 2025-12-23

**Authors:** Silvia Espinoza Barreiro, Diana Narvaez, Alhena Alfaro-Urquiola, Venus Medina-Maldonado

**Affiliations:** 1Gender-Based Violence Prevention Research Group (E-Previo), Faculty of Health and Wellbeing, Pontifical Catholic University of Ecuador, Quito 170143, Ecuador; sgespinozab@pucesd.edu.ec (S.E.B.); denarvaezb@pucesm.edu.ec (D.N.); 2Specialization’s Program in Family and Community Health Nursing, Pontifical Catholic University of Ecuador, Santo Domingo 230203, Ecuador; 3Department of Nursing, Pontifical Catholic University of Ecuador, Manabí 130801, Ecuador; 4Department of Psychology, Oviedo University, 33009 Oviedo, Spain; uo298006@uniovi.es; 5Center for Health Research in Latin America (CISeAL), Faculty of Health and Wellbeing, Pontifical Catholic University of Ecuador, Quito 170143, Ecuador

**Keywords:** adolescence, dating violence, mixed-methods, perceived motives, gender norms

## Abstract

Teen dating violence is a serious issue that affects the physical, psychological, and emotional well-being of adolescents. The purpose of this study was to examine the predominant perceived motives adolescents attribute to dating violence through the integration of quantitative and qualitative data. Methods: A concurrent mixed-methods design with equal weighting was applied to a sample of 703 participants in the quantitative phase, who completed the Dating Violence Motives Scale, and 103 participants in the qualitative phase. The mixed-phase analysis included data triangulation, creation of new analytical categories, and interpretation to generate meta-inferences. Results: Jealousy emerged as the most frequently perceived motive, particularly among males, followed by motives related to anger expression and lack of communication skills. Qualitative findings additionally revealed contextual elements not captured by the scale, such as family interference, relational control, and circumstantial stressors (academic, work, financial) as perceived triggers of violent behavior. Conclusions: Sociocultural constructions of gender were reflected in different motivational patterns: males more frequently justified violence as reactive or control-based, whereas females framed it as emotionally expressive.

## 1. Introduction

Teen dating violence (TDV) represents a serious public health concern that compromises the physical, psychological, and emotional well-being of those who experience it. This phenomenon manifests through behaviors that include physical, psychological, sexual, and economic violence in non-marital romantic relationships ([33]). According to the World Health Organization (WHO), approximately one in three women and one in five men experience some form of intimate partner violence, underscoring its classification as a public health problem with deep cultural and social roots ([40]).

Analyzing this phenomenon requires a multicausal approach that considers both the nature and the dynamic process of violent behavior, aiming to achieve a comprehensive understanding of its adverse outcomes, namely, diminished well-being and increased revictimization ([33]). Adolescence represents a critical period for the development of relational patterns that often persist in adulthood, making research and early prevention particularly relevant. It is also a developmental stage in which both risk and protective factors simultaneously interact to shape vulnerability to abusive relational dynamics, reinforcing the relevance of examining perceived motives during this period ([37]).

Accordingly, this study is grounded in the WHO’s ecological model, which posits that the determinants of intimate partner violence operate in an interrelated manner across individual, relational, community, and societal levels. This framework facilitates an understanding of why certain groups face greater risk than others ([38]). It also draws on the theory of coercive control, which holds that violence does not always manifest physically but may emerge as a systematic pattern of domination through partner-imposed isolation and surveillance, progressively restricting the victim’s autonomy ([17]; [36]). Such control is grounded in constant threat and emotional manipulation, potentially escalating into more severe forms of violence ([21]) without the victim’s awareness, as multiple risk factors operate simultaneously across different levels ([8]).

International research has identified diverse underlying motives contributing to TDV. For the purposes of this study, these have been classified into attitudinal factors, such as the tolerance and normalization of violence and the tendency to blame the victim ([1]), and exposure-related factors, including early experiences of intrafamilial violence and adherence to traditional gender norms ([14]). Adverse childhood experiences also play a significant role ([29]). Other contributing factors to TDV perpetration and victimization include alcohol and substance use ([6]), mental health and personality ([19]), social and peer influences, such as the adoption of inappropriate behaviors including harassment, coercive control, and violence ([7]), and relationship dynamics ([16]; [26]).

In Latin America, TDV exhibits characteristics shaped by the region’s sociocultural context. A previous study documented alarming rates of physical and/or sexual violence perpetrated by intimate partners, with prevalence estimates ranging from 14 to 17% in countries such as Brazil, Panama, and Uruguay to over 58% in Bolivia ([3]). These figures highlight the magnitude of the problem in a region marked by pronounced gender inequalities and a deeply entrenched culture of machismo. Structural factors such as poverty, social inequality, limited access to comprehensive sex education, and the weakness of victim protection systems exacerbate the issue. The intersection of gender-based violence, ethnic discrimination, and socioeconomic inequality creates heightened vulnerability for certain population groups.

In Ecuador, TDV constitutes a pressing concern that reflects both commonalities with the broader Latin American context and specific national features. A recent study identified links between self-esteem, coping strategies, and dating violence among Ecuadorian adolescents, illustrating how individual factors interact with sociocultural conditions particular to the country ([25]). Furthermore, research has identified social and cultural barriers that hinder help-seeking and coping processes in TDV situations among adolescents ([27]), including strong adherence to traditional gender norms, limited conflict resolution skills, and a lack of early identification of abusive behaviors, all of which impair emotional development ([20]).

Despite the growing body of literature on TDV, a knowledge gap remains. To the best of our knowledge, recent studies have not examined the specific situations that motivate violence in adolescent romantic relationships by integrating both objective data and subjective perceptions through a mixed-methods approach. Most existing research has focused on prevalence, violence typologies, or general risk factors, without systematically identifying the concrete motives or comparing the motives by sex.

In this study, the construct motives are conceptualized exclusively as perceived motives adolescents attribute to dating violence, rather than objectively verified causes or lived perpetration/victimization experiences. This conceptual distinction is central to the theoretical framing and interpretation of the findings in this research.

The aim of this mixed-methods study was to identify the predominant perceived motives underlying dating violence among adolescents by analyzing the quantitative results from a measurement instrument, and to complement these findings with qualitative insights that explored additional perceived motives not reflected in the survey but considered by participants as more representative of their reality. This approach allowed us to capture the subjective perceptions, meanings, and explanations that adolescents attribute to their behaviors and relational dynamics, thereby generating evidence to inform culturally adapted, context-specific preventive interventions.

## 2. Materials and Methods

This was a concurrent mixed-methods study, as both methods (QUAN + QUAL) were applied simultaneously with equal weighting throughout the research process ([35]). The study was conducted between September 2024 and January 2025 in seven different provinces of Ecuador. The target population consisted of adolescents, who represented 17.9 million people, equivalent to 33% of the total population according to the national census conducted in Ecuador in 2024 ([15]). Recruitment was carried out through the Adolescent Health Program in various primary care centers and school settings.

The eligibility criteria included adolescents between 14 and 19 years old who were currently in a dating relationship; willingness to participate; for minors, prior written informed consent from parents or legal guardians; and the adolescent’s signed assent. Participants who reported cohabitation with their partner were excluded from the study.

### 2.1. Quantitative Phase

A non-probabilistic purposive sampling method was used due to the difficulty of randomizing cases given the inclusion criteria requirements. We initially obtained agreement to participate from 732 adolescents. Cases were excluded for lack of age representativeness among 12-year-olds (3 cases) and for incomplete questionnaire responses (26 cases). The final response rate was 96.0%, resulting in a total of 703 adolescents included in the study analyses. Although the sample included adolescents from multiple provinces and diverse contexts in Ecuador, it was obtained through non-probabilistic sampling procedures and therefore should not be considered nationally representative. The findings should be interpreted with these limits of generalizability in mind. Our article presents results from the research project “Adolescent and University Student Dating Violence: A Dyadic Study”, funded by the Pontifical Catholic University of Ecuador.

The sample included 356 male adolescents (50.6%) and 347 female adolescents (49.4%), providing balanced representation of both sexes. Regarding age, 54.9% were in early adolescence (14–16 years), while 45.1% were in late adolescence (17–19 years). Most participants in both sexes self-identified as Mestizo. With respect to the family economic level, the responses were predominantly concentrated in the middle-class category, followed by the high-income category (see Table 1).

The primary variable of this study was the set of perceived motives adolescents attributed as triggers of dating violence. In this study, the term “perceived motives” refers to the situational trigger report by adolescents to justify or account for dating violence.

The scale asked adolescents to report the perceived motives that could lead them to perpetrate dating violence within their own current romantic relationships, rather than general beliefs about dating violence in society. Perceived motives were assessed using an abbreviated 13-item version adapted for adolescents from the motives framework proposed by [18] ([18]), evaluating the following domains: expression of anger, communication problems, demonstration of superiority, partner control, response to previous violence, self-defense, response to emotional harm, rage, punishment, proof of affection, sexual arousal, attention seeking, and jealousy. Each item was rated on a five-point Likert scale (Never, Sometimes, Frequently, Usually, Almost always).

The global internal consistency of the 13-item adapted scale in this adolescent sample was adequate (Coefficient ω = 0.829; Cronbach’s α = 0.849), supporting the reliability of the instrument.

Trained researchers conducted data collection to ensure consistent engagement with the participating institutions. Following institutional approval, the study was presented to adolescents, and those interested received informed consent forms for parental or legal guardian signature. On confirmation of parental consent, the adolescents signed informed assent forms, after which the instruments were administered. Anonymity and confidentiality were maintained throughout.

The quantitative analysis included descriptive statistics to summarize the participant demographics and to report the frequency and percentage of each motive. For descriptive analyses, the five response categories were collapsed into three levels (Never/Sometimes/Almost Always) to enhance interpretability, without modifying the conceptual meaning of the perceived motive variable. Ordinal logistic regression was used to calculate the odds ratios (OR) with 95% confidence intervals (CI) to assess the associations between each motive and sex, using females as the reference category. Due to insufficient variability and missing data in socioeconomic level and ethnicity variables, these indicators were not included in the inferential models, and sex was used as the primary grouping variable for comparative analysis.

### 2.2. Qualitative Phase

The qualitative component comprised written open-ended responses aimed at exploring adolescents’ perceptions and experiences, as well as identifying additional motives not included in the scale. This method was selected for its ability to generate rich textual data free from interviewer bias, while preserving anonymity and encouraging reflection. Participants were asked the following guiding question: In your opinion, what other perceived motives could lead you or your partner to perpetrate aggression in your current dating relationship? The open-ended question allowed unrestricted responses without suggesting content. As noted by [28] ([28]), the qualitative strength of this method lies in interpreting the meanings participants assign to their reported motives. A total of 103 out of the 703 participants voluntarily provided written qualitative responses.

Data were analyzed thematically following [4]’s ([4]) six-step framework: familiarization with the data, generation of initial codes, theme searching, theme review, theme definition and naming, and report production. An inductive logic guided theme development, enabling the identification of recurring patterns and categories linked to adolescents’ perceived motives. This process was carried out manually using analytic matrices developed in Excel and Word, rather than qualitative analysis software, which allowed iterative comparison, refinement, and consolidation of themes.

### 2.3. Mixed Phase

After analyzing each dataset according to the principles of its respective methodological paradigm, the findings were integrated to provide a more comprehensive understanding of the phenomenon under study ([35]). The purpose of mixing in this research was to achieve both breadth (through quantitative descriptive patterns) and depth (through qualitative explanatory meanings). Although the quantitative sample was larger, the study adopted an equal-status concurrent mixed-methods design, in which both strands were given equal interpretive weight at the meta-inference level. The primary point of integration occurred during the interpretation stage, where both strands contributed jointly to the generation of integrated conclusions ([35]). The validation strategy involved data triangulation, the creation of new analytical categories, and interpretative synthesis. This process facilitated the development of meta-inferences, representing the final interpretation of the results, and ensured the rigor and quality of the theorization process. The analyses were guided by the WHO ecological model ([8]; [38]) and the theory of coercive control ([17]; [21]; [36]).

### 2.4. Ethical Considerations

Participants were fully informed about the aims of the study, with emphasis on its voluntary nature and the guarantee of confidentiality. For minors, written informed consent was obtained from parents or legal guardians. All procedures strictly adhered to the ethical principles set forth in the Declaration of Helsinki, which serves as a framework for ethical standards in research involving human participants ([9]). This research project received formal approval from the Human Research Ethics Committee of the Pontificia Universidad Católica del Ecuador, under registration code PV-14-2022.

## 3. Results

### 3.1. Quantitative Phase

Table 2 summarizes the analysis of the perceived motives that adolescents attribute to dating violence, stratified by sex, in a sample of 703 Ecuadorian participants (50.6% male; 49.4% female). Percentual differences between males and females were observed. For females, the most frequent perceived motives were showing anger (7.1% vs. 5.3% “Almost always”), without words (6.5% vs. 5.7%), and demonstrate superiority (2.7% vs. 2.1%). For males, the leading perceived motives were because he/she hit me first (2.3% vs. 0.7%), to protect myself (3.4% vs. 2.8%), and to prove that he/she loved me (2.6% vs. 2.0%). The perceived motive “sexually arousing” showed the most notable sex gap (“Sometimes”: 6.3% vs. 2.4%), suggesting differences in the eroticization of violent behaviors. Jealousy was the most frequently perceived motive for both sexes, but with greater intensity in males (7.1% vs. 4.8% “Almost always”). Additional differences emerged for the perceived motive “punish my partner” (2.7% in females vs. 1.6% in males) and the perceived motive “call/attract his/her attention” (3.4% in males vs. 2.1% in females). These findings confirm meaningful sex-based differences in the perceived motives associated with dating violence.

Table 3 presents the data related to the perceived motives. Only one perceived motive showed a statistically significant association with sex: the item “sexually arousing” (*p* = 0.004), in which women were less likely than men to report this motive (OR = 0.474). Another perceived motive that draws attention, although not statistically significant, is “because he hit me first” (OR = 1.20), reported more frequently by women compared with men. In the case of jealousy, women were less likely to report this perceived motive. In contrast, the perceived motive “without words” was more frequently reported by women. However, these differences were not statistically significant (*p* > 0.05).

### 3.2. Qualitative Phase

This section presents the open-ended responses obtained from the qualitative component, which facilitated the exploration of adolescents’ own perceptions, meanings, and experiences.

#### 3.2.1. Self-Related Factors

Within this theme, which emerged as one of the most frequently referenced categories, adolescents perceived personal factors as influencing their relational dynamics and explaining why violent behaviors might occur. This category ranges from personal insecurities and jealousy to perceptions of infidelity, disrespect, and the need for recognition, reflecting the emotional turbulence, characteristic of adolescence, and its impact on romantic relationships.

PVNM-014A: “For flirting with other girls” (female);VNAC0001B: “For ignoring me about things he forbids me to do” (female);PVNQC-014A: “Comparisons with other partners” (female);PVNQC-004A: “For telling lies” (female);PVNQC-003B: “For avoiding me” (male).

#### 3.2.2. Lack of Communication Skills

This theme was also commonly mentioned by participants and refers to problems related to the inability to express emotions and resolve conflicts constructively. Adolescents noted that misunderstandings and inadequate communication generate conflicts that can quickly escalate into violent behaviors, especially when there is difficulty in clearly conveying ideas and feelings. Differences in ways of thinking and the inability to accept and manage these differences in opinion were mentioned as factors that may generate conflict within the relationship. Participants also identified the use of inappropriate jokes or remarks as a form of covert humiliation that undermines the relationship. The lack of emotional presence from one partner creates dissatisfaction that can lead to aggressive behaviors aimed at gaining attention, while unmet expectations regarding the relationship, such as a lack of punctuality or limited shared time, were cited as situational triggers for frustration that sparked conflict between partners.

PVNQC-003B: “Not answering me when I was calling her” (male);PVNP-049B: “Because she replied dryly in the chat” (male);PVNEC-025A: “Due to lack of communication” (female);PVNEC-014A: “For lack of dedication or time to spend together” (female).

#### 3.2.3. Family and Control

Although this theme was less frequently mentioned than the previous categories, adolescents still attributed relational conflict to situations involving family interference and the reproduction of dysfunctional relational patterns. Participants identified family interference as a factor that generated significant tensions, such as disapproval of the relationship or excessive attempts at control by family members. Imposed prohibitions, particularly those perceived as unfair or arbitrary, were cited as adding pressure to the relationship and potentially leading to reactive violent behaviors. Adolescents also highlighted how family conflicts experienced at home tend to be reproduced in their own relationships, making evident the principle of the intergenerational transmission of dysfunctional relational patterns. This category underscores how the family context not only sets models for interaction that adolescents replicate but can also directly intervene in the couple’s dynamic, creating additional conflicts.

PVNP-035B: “Because I don’t like it when she controls me” (male);VNT0110A: “Because relatives interfere in the relationship” (female);PVNP-007A: “Because of gossip from friends and especially family” (female).

#### 3.2.4. Circumstantial Problems

This final theme was mentioned by a minority of participants and encompasses external factors that significantly influence relationship dynamics and can precipitate violent episodes. Adolescents reported that academic or work-related pressures create stress that often spills over into the relationship, fostering an environment conducive to conflict. Economic difficulties and financial problems were identified as major sources of tension that, if not managed effectively, can lead to violent confrontations. Although these factors are external to the relationship itself, they were recognized as increasing the couple’s vulnerability to violent dynamics by raising general stress levels and reducing the capacity to constructively manage disagreements.

VNL0106B: “For lack of money” (male);PVNP-007A: “For having some financial problems” (female);PVNEC-024A: “Because I have had problems at work” (female).

### 3.3. Integration of the Results

Table 4 functions as a joint display, presenting a mixed-methods matrix that integrates the quantitative descriptive patterns with the qualitative thematic explanations, highlighting points of convergence and divergence across strands. An analysis of the convergences yielded the first meta-inference: jealousy was the most frequently perceived situational trigger associated with dating violence. This was identified both in the quantitative variables and in the narratives from the qualitative phase.

The second meta-inference, also emerging from the similarities reported in both phases of the study, referred to sex-differentiated motivations. In males, this motive was mainly linked to controlling behaviors toward the partner, whereas in females it was expressed as an emotional response to situations perceived as threatening to the relationship, a finding consistent with previous studies on sex-specific patterns.

The third meta-inference identified internal individual factors present in both study phases and related to deficits in the communication and emotional management skills necessary for negotiation processes aimed at healthy relationships. These perceived limitations may help explain why some adolescents’ attribute violence to difficulties in emotional regulation and communication, rather than to intentional aggression. The inability to express emotions and resolve conflicts verbally may foster the use of violent behaviors as an alternative form of communication, suggesting the need to incorporate the development of socioemotional and communication skills as an essential component in preventive programs.

Exclusively in the qualitative phase, meta-inferences emerged related to relational factors, referring to family influence (interference, prohibitions, or conflicts), and external sociocultural factors, such as work, academic, and financial problems, as well as conflicts with the immediate social environment. These elements, not included in the quantitative instrument, enrich the understanding of the phenomenon by incorporating specific contextual and cultural dimensions.

The integration of these findings indicates that adolescent dating violence results from the interaction between individual, relational, and contextual dimensions, reinforcing the relevance of designing preventive approaches with a multicausal focus. Moreover, the qualitative responses revealed cultural nuances and specificities not fully addressed in the questionnaire, suggesting the potential for developing culturally adapted assessment tools that more accurately reflect the dynamics and challenges specific to Ecuadorian adolescents regarding the motives for dating violence.

In the integrated interpretation, the quantitative results primarily provided a descriptive mapping of the relative prominence of perceived motives across sex, whereas the qualitative phase offered explanatory depth regarding how adolescents interpret and give meaning to these motives. Therefore, sex-related interpretations must be viewed cautiously, as most differences were descriptive rather than inferentially supported, and only one perceived motive showed a statistically significant association.

## 4. Discussion

The main objective of this research was to integrate quantitative scale findings with qualitative textual narratives to broaden the understanding of the perceived motives that adolescents attribute to dating violence. These results should be interpreted as subjective self-reported explanations rather than objective causal mechanisms of violent behavior. Consistent with other mixed-methods studies ([20]), the theoretical frameworks used in this study supported the generation of meta-inferences and facilitated the interpretation of how adolescents make sense of and justify these behaviors within their relational contexts.

The integrated results of the quantitative and qualitative phases show that jealousy was the most frequently *perceived* cross-cutting motive associated with dating violence, with greater intensity reported by males. We also identified sex-differentiated patterns: in males, motives associated with control and domination predominated, whereas in females, emotional responses to perceived threats to the relationship were more frequent. Other relevant perceived motives included lack of communication skills, family-related perceptions, and circumstantial factors.

Our findings are consistent with the previous research, highlighting that traditional values about romantic love, particularly those that idealize jealousy as a sign of love, significantly predict both the perpetration and victimization of violence ([13]). In our study, this motive appeared with greater intensity among males, consistent with the research showing that 68.4% of adolescents consider jealousy to be proof of love, with this belief more frequent in men (72.3%) than in women (64.5%) ([31]). Likewise, the literature indicates that in boys, jealousy is mainly linked to the normalization of violence ([11]) and exposure to domestic violence, whereas in girls, it is more associated with low emotional regulation ([22]). Another study has also identified jealousy as a trigger for violent behavior, with cycles of aggression often beginning with arguments in the context of interactions through social networks ([32]). Taken together, these findings illustrate that adolescents often interpret jealousy as a sign of love, which may contribute to its perceived normalization within romantic relationships.

Another relevant finding was the sex differences observed in this study, which are in line with prior research showing that males tend to display attitudes supportive of violence and controlling behaviors, whereas low emotional regulation is more common in females ([11]; [30]; [31]). In this regard, one study reported higher perpetration rates among men (84.6% versus 15.4% in women) and higher victimization rates among women (87.6% versus 12.4% in men) ([10]). Another study found that emotional control and manipulation constitute the first link in the escalation of violence in adolescent relationships ([34]). The consistency of these patterns across different cultural contexts suggests the existence of a cross-cultural pattern in the differentiated manifestations of violence by sex. This finding reflects deeply rooted sociocultural constructions of gender in different societies, even when the motives for perpetrating dating violence are not directly explored. However, these sex-related patterns should be interpreted with caution, as inferential support was limited and most differences were descriptive rather than statistically significant.

Regarding the identification of deficits in communication skills, a study by [27] ([27]) corroborated the importance of this factor, noting that the ability to express emotions and negotiate conflicts assertively significantly reduces the likelihood of resorting to violent behavior. In our study, the convergence between quantitative data, where without words was identified as a significant motive (6.5% in women and 5.7% in men), and the thematic unit “Lack of communication skills” validates this dimension as a cross-cutting factor in the generation of violence ([36]).

From a theoretical perspective, these findings can be interpreted in light of the WHO ecological model ([38]). Within this framework, individual factors include insecurity, jealousy, and lack of communication skills; relational factors encompass family conflicts and controlling behaviors toward adolescents; and sociocultural factors include the academic and work realities of adolescence, as well as the economic difficulties present in these societies. All these elements illustrate the multilevel conditions that adolescents perceive as contributing to dating violence and that should be considered equitably in prevention programs. No single factor is more important than another, as all shape the meanings adolescents attribute to their romantic relationships. In this regard, coercive control theory provides a framework for interpreting male perceived motives, while the gender perspective ([41]) explains differences in meanings and expressions according to sex. Therefore, this study contributes to understanding how adolescents rationalize and make sense of dating violence, rather than establishing direct causal pathways.

In the Latin American context, beliefs surrounding romantic love often normalize jealousy as an expected and even desirable expression of attachment, emotional exclusivity, and validation of the relationship ([25]). Within Ecuador, this belief is reinforced by broader cultural expectations rooted in machismo and gendered scripts in which emotional control, possessiveness, and vigilance are interpreted as proof of commitment ([27]). The cultural narratives contribute to the normalization of jealousy-based violence and help explain why adolescents do not always interpret jealousy as a risk signal, but rather as an expected component of romantic involvement. This cultural meaning-making strengthens the interpretation of jealousy as a perceived motive across both strands of data in this study.

### 4.1. Practical Implications

This study is particularly relevant to the health and education sectors. In the field of community health, our approach enabled the integration of observable behaviors with the experiences and meanings that adolescents themselves attribute to their actions and relational dynamics, thereby generating evidence to inform the design of culturally adapted interventions tailored to local realities. Within the framework of SDG 3 on Health and Well-being, intersectoral collaboration is a key strategy, opening the possibility of implementing school-based programs that address these issues in line with the principles of promoting equality and comprehensive health, as advocated by the WHO and the 2030 Agenda, to strengthen the well-being of adolescents as a vulnerable group ([39]).

Regarding intimacy and sexual scripts, the findings reveal a notable pattern in the motivations underlying adolescent dating violence, particularly in relation to gender differences. It is significant that the motive described as “sexually arousing” was reported by boys, which highlights the need to address cognitive distortions about consent and intimacy, as well as dysfunctional associations between sexual violence, intimacy, and aggressive sexual scripts in psychotherapeutic interventions with adolescent males. From a forensic perspective, this can also be considered a specific risk factor associated with the profile of male aggressors, which should be taken into account in dangerousness assessments and in judicial proceedings, where motivational patterns are examined ([5]; [12]; [24]).

Conversely, the absence of significant differences in other perceived motives, such as control, emotional response, or punishment, suggests a relative sex symmetry in the motives adolescents report regarding dating violence. This challenges certain stereotypes and underscores the importance of context-sensitive clinical approaches that are not exclusively focused on sex and gender. Similarly, the presence of perceived motives such as “to prove that he loved me,” “to get his attention,” or “without words,” although not statistically significant, highlights the need to address dysfunctional affective patterns, insecure attachment styles, and myths of romantic love that adolescents associate with violent behavior ([2]; [23]).

These results call for a rethinking of preventive interventions, incorporating a more nuanced understanding of motivation, attention-seeking, jealousy, and emotional impulsivity, regardless of the aggressor’s sex. This requires implementing affective education, cognitive restructuring, and emotional regulation strategies from early stages of development. As for future research, it is essential to validate culturally adapted instruments for measuring the motivations for violence in adolescents and to develop studies involving adolescent couples, enabling the evaluation of how both the perpetrator and their partner contribute to the dynamics of the relationship.

Although protective factors did not emerge explicitly in this study, future research should examine elements such as peer support, emotional regulation competencies and conflict resolution skills, which could contribute to a more balanced, strengths-based approach to prevention of adolescent dating violence.

### 4.2. Limitations

The cross-sectional design prevented the establishment of causal relationships. The non-probabilistic sampling and reliance on self-reported data introduce biases, which are factors that limit the generalizability of the findings to the entire adolescent population in Ecuador. Sociodemographic variables such as ethnicity and socioeconomic level were not included in the inferential modeling due to insufficient variability and missing data, which limits the ability to explore their contribution to perceived motives. Nevertheless, the large sample size, data collection across multiple provinces of the country, and the mixed-methods design strengthen the relevance of the results. Additionally, the instrument was originally developed for a university population and to identify motives for the perpetration of physical violence. This, together with potential cultural differences, may also affect the generalizability of the findings.

## 5. Conclusions

This concurrent mixed-methods study on the motives for adolescent dating violence in Ecuador reveals that psychological violence is the predominant form of aggression, with motivational differences observed by sex and stage of adolescence. Jealousy emerged as the most prevalent cross-cutting motive in both sexes, being more pronounced among males, reflecting the persistence of distorted beliefs about romantic love in the Ecuadorian context

Women tended to engage in violence for expressive–emotional perceived motives, whereas men did so for reactive or control-oriented perceived motives. A lack of communication skills was identified as a critical factor in both the quantitative and qualitative phases, underscoring the need to develop competencies for assertive expression and peaceful conflict resolution. The influence of familial and sociocultural factors was evident in the qualitative phase, illustrating how dysfunctional relational patterns are transferred across contexts. These findings underscore the need for culturally adapted preventive interventions from all institutions involved in adolescent development, whether in health or education, employing a differentiated approach based on sex and stage of adolescence. Such interventions should prioritize the denormalization of jealousy as an expression of love, the development of communication skills, and the transformation of cultural beliefs about emotional relationships, thereby contributing to the creation of healthy, equitable, and violence-free bonds among Ecuadorian adolescents.

## Figures and Tables

**Table 1 behavsci-16-00031-t001:** Demographic characteristics of the study participants (*n* = 703).

Variables	Total		Early Adolescence		Late Adolescence	
Sex	*n*	(%)	*n*	(%)	*n*	(%)
Male	356	(50.6)	195	(27.7)	161	(22.9)
Female	347	(49.4)	191	(27.2)	156	(22.2)
Economic level						
Low	106	(15.1)	58	(8.3)	48	(6.8)
Medium	456	(64.9)	251	(35.7)	205	(29.2)
High	141	(20.1)	77	(11.0)	64	(9.1)
Ethnicity						
Mestizo	627	(89.2)	344	(48.9)	283	(40.3)
Afro-descendant	30	(4.3)	17	(2.4)	13	(1.9)
Indigenous	30	(4.3)	17	(2.4)	13	(1.9)

Source: Data obtained from the survey.

**Table 2 behavsci-16-00031-t002:** Descriptive statistics on the perceived motives for violence by sex.

Sex					
Perceived Motives	Behavior Frequency	Male *f* (%)	Female *f* (%)	Total	Range
Show anger	Never	197 (28)	161(22.9)	358	1–3
	Sometimes	122 (17.4)	136 (19.3)	258	1–3
	Almost always	37 (5.3)	**50 (7.1)**	87	1–3
Without words	Never	202 (28.7)	188 (26.7)	390	1–3
	Sometimes	114 (16.2)	113 (16.1)	227	1–3
	Almost always	40 (5.7)	**46 (6.5)**	86	1–3
Demonstrate superiority	Never	274 (39.0)	270 (38.4)	544	1–3
	Sometimes	67 (9.5)	58 (8.3)	125	1–3
	Almost always	15 (2.1)	19 (2.7)	34	1–3
Control my partner	Never	261 (37.1)	270 (38.4)	531	1–3
	Sometimes	74 (10.5)	55 (7.8)	129	1–3
	Almost always	21 (3.0)	22 (3.1)	43	1–3
Because he/she hit me first	Never	303 (43.1)	302 (43.0)	605	1–3
	Sometimes	37 (5.3)	40 (5.7)	77	1–3
	Almost always	**16 (2.3)**	5 (0.7)	21	1–3
To protect myself	Never	287 (40.8)	282 (40.1)	569	1–3
	Sometimes	45 (6.4)	45 (6.4)	90	1–3
	Almost always	**24 (3.4)**	20 (2.8)	44	1–3
Response to emotional harm	Never	274 (39.0)	262 (37.3)	536	1–3
	Sometimes	57 (8.1)	56 (8.0)	113	1–3
	Almost always	25 (3.6)	**29 (4.1)**	54	1–3
Out of anger	Never	272 (38.7)	249 (35.4)	521	1–3
	Sometimes	65 (9.2)	79 (11.2)	144	1–3
	Almost always	19 (2.7)	19 (2.7)	38	1–3
Punish my partner	Never	302 (43.0)	283 (40.3)	585	1–3
	Sometimes	43 (6.1)	45 (6.4)	88	1–3
	Almost always	11 (1.6)	19 (2.7)	30	1–3
To prove that he/she loved me	Never	263 (37.4)	271 (38.5)	534	1–3
	Sometimes	75 (10.7)	62 (8.8)	137	1–3
	Almost always	**18 (2.6)**	14 (2.0)	32	1–3
Sexually arousing	Never	305 (43.4)	322 (45.8)	627	1–3
	Sometimes	44 (6.3)	17 (2.4)	61	1–3
	Almost always	7 (1.0)	8 (1.1)	15	1–3
To call/attract his/her attention	Never	261 (37.1)	269 (38.3)	530	1–3
	Sometimes	71 (10.1)	63 (9.0)	134	1–3
	Almost always	24 (3.4)	15 (2.1)	39	1–3
Out of jealousy	Never	211 (30.0)	215 (30.6)	426	1–3
	Sometimes	95 (13.5)	98 (13.9)	193	1–3
	Almost always	**50 (7.1)**	34 (4.8)	84	1–3

**Source:** Data obtained from the survey. *Note.* Values represent the frequency with which adolescents *perceive* each motive could lead them to perpetrate dating violence; these are not behavioral frequencies. The original five-point Likert scale was collapsed into three categories for descriptive purposes (Never/Sometimes/Almost Always). Note. Bold indicates the highest frequency within each perceived motive across sex. No statistical significance is implied.

**Table 3 behavsci-16-00031-t003:** Ordinal logistic regression for each motive by sex.

Perceived Motives	OR (Women vs. Men)	IC95% Lower	IC95% Upper	*p*-Value
Show anger	0.888	0.666	1.18	0.417
Without words	0.901	0.669	1.21	0.494
Demonstrate superiority	0.971	0.683	1.38	0.871
Control my partner	0.802	0.570	1.13	0.208
Because he/she hit me first	**1.20**	0.789	1.85	0.383
To protect myself	0.950	0.653	1.38	0.789
Response to emotional harm	0.913	0.647	1.29	0.607
Out of anger	0.796	0.569	1.11	0.184
Punish my partner	0.777	0.523	1.15	0.212
To prove that he/she loved me	0.793	0.561	1.12	0.188
Sexually arousing	0.474	0.287	0.784	*** 0.004**
To call/attract his/her attention	0.784	0.557	1.10	0.163
Out of jealousy	0.852	0.634	1.14	0.286

***Note:*** OR > 1 indicates a higher likelihood for women; OR < 1 indicates a lower likelihood for women. Prevalence ratio refers to the frequency with which the motive occurs. Note. Bold indicates statistically significant results (*p* < 0.05). An asterisk (*) denotes statistically significant results (*p* < 0.05).

**Table 4 behavsci-16-00031-t004:** Data matrix comparing the main quantitative, qualitative, and meta-inference results.

Quantitative Phase	Theme	Predominant Motives Qualitative Phase	Meta-Inferences
Jealousy ↑ (20.60%) **Men** Showing anger ↑ (26.4%) **Women**	Self-related factors	Personal insecurities Jealousy Infidelity (real or perceived) Disrespect Lies Emotional manipulation Comparisons Lack of interest in the relationship Lack of emotional support Need for recognition	**Jealousy is the main motive for violence:** This motive appears in both the quantitative and qualitative data. **Sex-differentiated motives:** Men more frequently perceived their motives as related to control, whereas women more frequently perceived emotional responses as explanations.
Without words ↑ (22.6%) **Women**	Lack of communication skills	Misunderstandings Poor communication Different ways of thinking Inappropriate jokes or remarks Lack of presence in the relationship Lack of time/punctuality Unmet expectations	**Internal factor** (individual)
Not reported in instrument	Family and control	Family interference Family-imposed restrictions Family and/or friendship conflicts	**Relational factor** (family)
Not reported in instrument	Circumstantial Problems	Work-related problems Academic problems Financial/money problems	**External factor** (sociocultural)

**Note.** The upward arrow (↑) indicates a higher prevalence or greater frequency of the motive within the specified sex in the quantitative phase. Percentages refer to the proportion of participants reporting the motive.

## Data Availability

The data presented in this study are available on request from the corresponding author.

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
