# Peer review of "Understanding Perceived Motives for Dating Violence Among Adolescents: A Mixed-Methods Approach"

_behavsci, 2025, doi:10.3390/bs16010031_

Round 1

Reviewer 1 Report

Comments and Suggestions for Authors

In the introduction, on p.1, the addition of references on adolescence could further strengthen the theoretical solidity of the contribution. It is information that is barely mentioned at this moment.

It would be beneficial to integrate recent studies, particularly in reference to WHO documents and the ecological model, to broaden and update the conceptual framework, starting from the 2012 WHO document (or others, such as Agueli et al., 2024).

The specific figure on Latin America, which represents added value and significantly enriches the discussion, is very appreciable.

The section dedicated to the method constitutes a central element to guarantee the transparency and replicability of the study. In this version, however, it requires some significant revisions. 
In particular, it would be appropriate to describe in greater detail the methods of recruiting participants.

Additionally, it would be relevant to specify the tools used, highlighting their main characteristics and the reasons for their choice, as well as clarifying how they were administered or adapted to the research context. It is not clear whether these are validated scales, individual items or ad hoc items.
The same applies to qualitative analysis; it would be helpful to integrate a more detailed description of the process through which the open questions were processed. 
An in-depth study of these aspects would not only make the section more complete but would also increase the methodological solidity and overall readability of the work.
It should also be specified whether data analysis software has been used, and if so, which and how.

The results of the qualitative part present margins of enrichment. Currently, the section appears concise and may not fully convey the complexity and depth of the experiences collected. It would be desirable not only to expand the use of direct quotations but also to accompany them with comments and interpretative additions that would help to contextualise them and bring out the underlying meanings. An expansion in this direction would contribute to increasing the interpretative richness and narrative strength of the section, making it more incisive and consistent with the objectives of the study.

The section dealing with the integration of results is well developed. The articulation is clear, allowing the different data to be effectively brought into dialogue and contributing to the overall coherence of the work. This passage represents a strong point of the manuscript, as it enhances the results and facilitates their interpretation.

The discussion section is overall well developed and in line with the objectives of the manuscript. However, it could be further strengthened through a more detailed study of the part relating to the ecological model. A more comprehensive reference to this theoretical framework, with specific citations to the most recent literature and a more detailed reflection on its implications for the results that emerged, would contribute to making the discussion more robust and enhancing its conceptual relevance.

Reviewer 2 Report

Comments and Suggestions for Authors

This study used a concurrent mixed-methods design to examine the reasons adolescents in Ecuador engage in dating violence, combining survey data from over 700 youth with open-ended qualitative responses from a subsample. Across both strands, jealousy emerged as the most frequent motive, alongside themes of poor communication skills, emotional responses, and contextual stressors such as family interference and financial strain. A notable strength of the study is using both quantitative and qualitative approaches, and I like that using both allowed breadth and depth in understanding motives. Another strength is the large and balanced sample across sex and age groups, so I commend the authors on that.

  • The study notes “equal weighting” of quantitative and qualitative strands, but the quantitative sample is much larger and dominates the analysis. The qualitative findings feel somewhat supplementary. Maybe include a stronger joint display or mixed-methods matrix to show more clearly how the two strands intersect and diverge?
  • More detail on how meta-inferences were generated (beyond listing overlapping themes) would strengthen the rigor of the integration.
  • The study highlights Latin American machismo and Ecuadorian sociocultural factors, but these are discussed mostly in the introduction and conclusion. A deeper thematic analysis of cultural beliefs about jealousy and love could be super helpful for the reader.
  • There is limited attention to protective factors (e.g., peer support, conflict resolution skills), which could provide a more balanced view for prevention programs. If those didn't come through, perhaps comment on why or future research could look into.

Reviewer 3 Report

Comments and Suggestions for Authors

This study addresses a relevant topic concerning adolescent perceptions of dating violence motives, using a mixed-methods approach. However, conceptual and methodological inconsistencies — particularly the conflation / confusion between perceptions and experiences, inconsistent terminology, and insufficient analytical rigor — limit the interpretability and impact of the findings. The paper would benefit from clearer conceptual framing and stronger justification for methodological and analytical choices.

Specifically,

The abstract highlights jealousy as the main motive attributed to dating violence; however, the result is overly generic and insufficiently informative. Similarly, the concluding statement “The findings indicate that sociocultural gender constructions influence …...” is too broad and lacks analytical depth. The discussion and implications should be more specific and linked to the empirical findings.

The introduction is generally well structured. However, the theoretical framing should clarify that the study concerns perceptions of motives for dating violence, rather than the motives themselves. This conceptual distinction is crucial, as the manuscript currently conflates the experience of victimization/perpetration with perceptions about victimization and perpetration in dating violence. The theoretical background should explicitly address this difference.

Concerning “Materials and Methods”, are sample characteristics representative of the national youth population? this information is crucial for the strength and generalizability of the (quantitative) results.

The inclusion of adolescents aged 14–19 who were currently in a dating relationship requires justification. Were participants asked about motives within their own relationships or about general perceptions of dating violence motives? This should be clearly stated.

Variables (Gender / Economic Level / Ethnicity): These sociodemographic variables are mentioned but not used in the analyses. Given the cultural and social relevance of these factors in shaping perceptions and motives for dating violence, their inclusion should be considered. The use of ordinal logistic regression only to assess associations between motives and sex is limiting. Why these other sociodemographic variables (e.g., gender identity, socioeconomic level, ethnicity) were not included?

The terms gender and sex are used interchangeably, which should be corrected.

The terms motives, reasons, and motivations are also used indistinctly. The manuscript should define the theoretical construct being measured and maintain terminological consistency. (e.g., reasons  - Table 2, and motivations - Table 3).

The description of the 13-item scale assessing motives (e.g., expression of anger, partner control, jealousy, etc.) lacks methodological detail. How was this scale constructed or validated? Furthermore, it is unclear whether the Likert scale (five-point) refers to personal experiences or perceived motives.

The qualitative procedure and guiding question(s) are not sufficiently clear. The manuscript should specify what question participants were asked and provide more information on who the 103 participants were.

The text mentions a five-point Likert scale, but Table 2 presents a three-point scale. Moreover, the table links “motives” with the frequency of behaviors — which conceptually differs. Are these items measuring behaviors motivated by specific motives or perceptions of motives? Again, the conceptual confusion between experiences and perceived motives needs clarification.

The statement “Significant gender differences were observed” (Table 2) is not supported, as only descriptive statistics are presented. Statistical tests must be conducted to substantiate claims of significance. Only statistically significant results should be reported.

The reporting of qualitative results would benefit from including the frequency of each category or theme identified.

The section on integration of findings is the most interesting part of the paper. However, the contribution of the quantitative phase to the overall meta-inferences is not entirely clear. The analytical strategy in the quantitative phase does not seem optimal for supporting the integrated interpretation — especially considering the absence of formal gender difference tests and the fact that the ordinal regression identified associations for only one motive. Greater caution should therefore be applied in interpreting gender-related patterns.
